

# A taxonomic outline of the *Poecilimon affinis* complex (Orthoptera) using the geometric morphometric approach

Maciej Kociński[1], Beata Grzywacz[1], Georgi Hristov[2] and Dragan Chobanov[2]

[1] Institute of Systematics and Evolution of Animals, Polish Academy of Sciences, Kraków, Poland
[2] Institute of Biodiversity and Ecosystem Research, Bulgarian Academy of Sciences, Sofia, Bulgaria

## ABSTRACT

The genus *Poecilimon* contains 145 species, widely distributed in the Palaearctic, among which the *Poecilimon ornatus* group has the greatest diversity in the Balkans. Despite several revisions of the genus, the systematics of the species group, and in particular, of the taxa associated with the species *Poecilimon affinis*, is still unsolved. Due to morphological similarity, *P. affinis* with its subspecies, *P. nonveilleri* and *P. pseudornatus* form the *Poecilimon affinis* complex. The aim of this study is to test the hypotheses of an outlined species complex, namely the *P. affinis* complex, within the *P. ornatus* group using morphological data. Geometric analysis was conducted to explore variation in the structure of the male tegmen, ovipositor, male cercus, and male pronotum. The number of teeth and stridulatory file measurements provided additional information on morphological variation within the complex. A phylogenetic tree based on the cytochrome c oxidase subunit I gene (COI) was used for comparison with the morphological data. Canonical variate analysis showed that male tegmen and male cercus are good morphostructures to distinguish the taxa belonging to the *P. affinis* complex from other species in the *P. ornatus* group. This may confirm our assumption for the designation of the *P. affinis* complex. The results of the principal component analysis of stridulatory file measurements, molecular data, and CVA of the ovipositor suggest adding two additional species to the complex: *P. ornatus* and *P. hoelzeli*.

## INTRODUCTION

*Poecilimon* Fischer, 1853 is one of the most species-rich genera within the Phaneropterinae subfamily. This genus comprises 145 species distributed in the Palearctic region (*Cigliano et al., 2021*). All species are short-winged and flightless herbivorous bush-crickets with complex acoustic behavior (*Heller, 1990*). *Poecilimon* is currently divided into 18 species groups based on molecular, morphological and bioacoustic data, while 16 species are not assigned to any of them (*Cigliano et al., 2021*). The similarity and variability of morphological characteristics make many *Poecilimon* species difficult to identify. The *Poecilimon ornatus* group (13 species and five subspecies) (Fig. 1) is one of the groups for which the phylogenetic relationships between species remain unclear and the status of several taxa is under discussion. Due to the reduced wings and the influence of climatic

Corresponding author
Maciej Kociński,
kocinski@isez.pan.krakow.pl

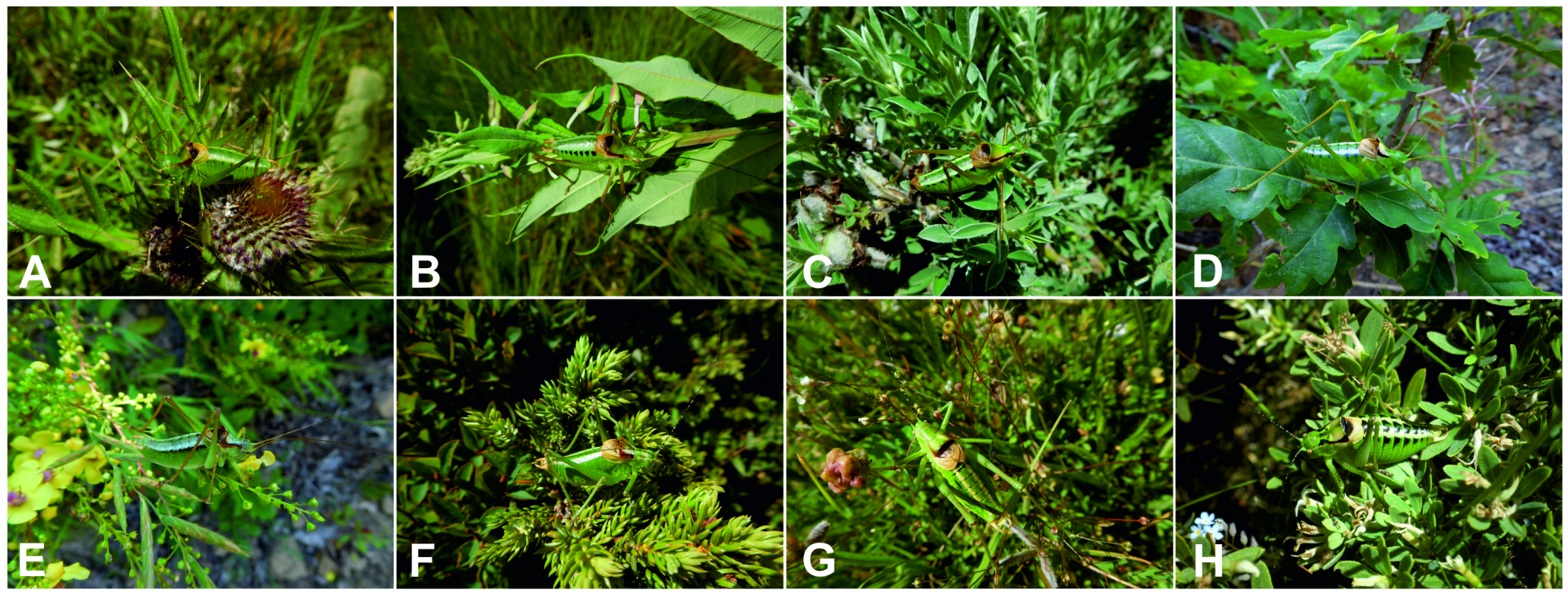

**Figure 1** **Representatives of the studied taxa from the *Poecilimon ornatus* group.** (A) *P. affinis hajlensis.* (B) *P. affinis affinis.* (C) *P. hoelzeli.* (D) *P. rumijae.* (E) *P. nonveilleri.* (F) *P. poecilus.* (G) *P. pseudornatus.* (H) *P. ornatus*. Photos: D. Chobanov.

and geomorphological factors, a rapid morphological evolution took place in this group (*Chobanov & Heller, 2010*).

The first revision of *Poecilimon* was conducted by *Ramme (1933)*, who included taxa from the currently recognized *Poecilimon ornatus* group in "Gruppe I." In *1984, Heller* suggested dividing the group into eight taxa (*P. nobilis* Brunner von Wattenwyl, 1878; *P. obesus obesus* Brunner von Wattenwyl, 1878; *P. obesus artedentatus* Heller, 1984; *P. affinis affinis* (Frivaldszky, 1867); *P. affinis komareki Cejchan, 1957*; *P. affinis hoelzeli* Harz, 1966; *P. ornatus* (Schmidt, 1850) and *P. pancici* Karaman, 1958; distributed mainly in the Balkans). Later, *P. artedentatus* and *P. hoelzeli* were given species status (*Willemse, 1985*; *Willemse & Heller, 1992*), while *P. pancici* was synonymized (*Willemse, 1985*). Further, six new species were described (*P. pindos* F. Willemse, 1982; *P. soulion* L. Willemse, 1987; *P. gracilioides* F. Willemse & Heller, 1992; *P. jablanicensis* Chobanov & Heller, 2010; *P. pseudornatus* Ingrisch & Pavicevic, 2010; *P. nonveilleri* Ingrisch & Pavicevic, 2010).

Among the *P. ornatus* group, *P. affinis* has the widest geographic range. It is distributed from northern Greece to the Carpathians in Romania and an isolated spot in Ukraine. According to *Cigliano et al. (2021)*, *P. affinis* consists of five subspecies (*P. affinis affinis* (Frivaldszky, 1868); *P. a. dinaricus* Ingrisch & Pavicevic, 2010; *P. a. hajlensis* Karaman, 1974; *P. a. komareki* Cejchan, 1957; *P. a. serbicus* Karaman, 1974). *Karaman (1974)* reduced the status of *P. poecilus* Ramme, 1951 to a subspecies of *P. affinis* and described two new subspecies: *P. a. serbicus* and *P. a. hajlensis*. In *1984, Heller* suggested that *P. poecilus* and *P. a. affinis* are synonymous. Due to doubts about the taxonomic status of *P. poecilus*, in the present study it will be treated separately. *Poecilimon komareki* was described by Cejchan (1957), but *Heller (1984)* regarded it as a subspecies of *P. affinis* because of their similarity. *Karaman (1972)* described *P. komareki rumijae* based on the shape of the male pronotum and body size. Because of the lowering of the status of *P. komareki* to a subspecies of *P. affinis*, *P. k. rumijae* became synonymous of *P. a. komareki*, which was confirmed by *Chobanov & Heller (2010)*. On the other hand, *Ingrisch & Pavicevic (2010)* suggested regarding *P. rumijae* as a separate species, differing distinctly from *P. affinis*.

Morphological variability in these taxa was determined only based on minor differences in the shape of the male pronotum and body size (*Chobanov & Heller, 2010*). Furthermore, song of *P. a. komareki* and *P. rumijae* resembles that of *P. pseudornatus* with a long silent beginning. Song of *P. nonveilleri* is short with a typical structure, whereas *P. a. affinis* has also short song and shows morphological differences to *P. nonveilleri* (own unpublished data). Due to the discrepancy between the authors, *P. rumijae* will also be treated separately in the present study. *Poecilimon pseudornatus*, *P. nonveilleri* and the subspecies of *P. affinis* are morphologically similar, although a recent molecular study based on the cytochrome c oxidase I gene has shown that the above taxa do not form a monophyletic group (*Kociński, 2020*). The lack of clear boundaries between them and the unsolved phylogenetic relationship suggest that *P. pseudornatus*, *P. nonveilleri* and subspecies of *P. affinis* should be treated as the *P. affinis* complex.

The 'species complex' is an informal taxonomic term showing the uncertainty of taxonomic identification (*Sigovini, Keppel & Tagliapietra, 2016*) and it is commonly used in insects (e.g., *Genier & Moretto, 2017*; *Manani et al., 2017*; *Elfekih et al., 2018*; *Selnekovič & Kodada, 2019*). It may be defined as a group of very closely-related taxa with similar morphology and difficult to distinguish from one another. Taxa from a complex require a critical revision in order to clarify the actual taxonomic position (*Sigovini, Keppel & Tagliapietra, 2016*).

To determine the morphological variation of the *Poecilimon ornatus* group, especially within the *Poecilimon affinis* complex, we used geometric morphometric methods based on the shape variation of four structures: male pronotum, male cercus, ovipositor, and male tegmen (Fig. 2). Geometric morphometrics is an approach that applies the landmark coordinates, which are the correspondence points marked on a given morphostructure and are the same in all studied specimens or species (*Bookstein, 1991*; *Dryden & Mardia, 1998*). This method considers the spatial relationships between landmark variables, therefore providing more powerful statistical results. It is also possible to find and analyze shape variations in the species within and between populations (*Walker & Bell, 2000*). The geometric morphometric method has been proved to be very useful for distinguishing species in insects (*Nunes et al., 2012*; *Prado-Silva et al., 2016*; *Da Silva et al., 2018*), especially in Orthoptera (*Romero, Rosetti & Remis, 2014*; *Barcebal et al., 2015*; *Kaya, Boztepe & Çiplak, 2015*; *Kaya et al., 2015*; *Mugleston et al., 2016*; *Bian & Shi, 2018*; *Pan, Hong & Jiang, 2018*; *Liu, Chen & Liu, 2020*). The aim of the present study is to assess the morphological diversity of the species within the *P. ornatus* group, outline morpho-units and discuss the importance of morphological traits for the systematics of the group. We test the hypothesis of the existence of the *P. affinis* complex.

## MATERIALS & METHODS

### Specimen collection

Bush-crickets were collected in the Balkan Peninsula (Bulgaria, Serbia, Montenegro, Albania, North Macedonia, Greece) between 2017 and 2019 and stored in 96% ethanol (Table 1). In Greece, field studies were approved by the Greek Ministry of the

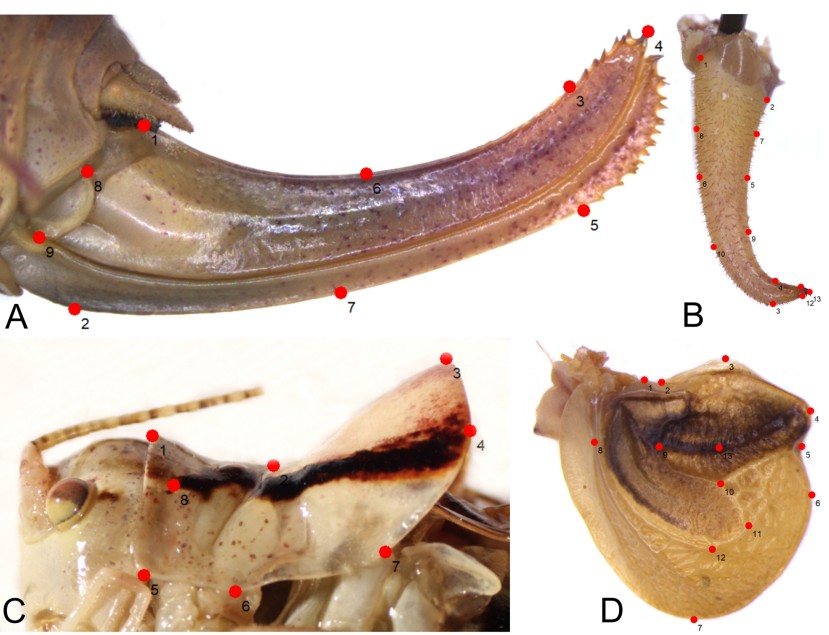

**Figure 2** **Position of the landmarks (red dots) on *Poecilimon* species used for geometric morphometrics.** (A) Ovipositor. (B) Male cercus. (C) Male pronotum. (D) Male tegmen.

Environmental, Energy, and Climate Change (No 154812/951). In Bulgaria, we did not need a permit for collecting for scientific purposes because it was outside protected areas, and animals were not protected. The material was collected with scientific purpose through scientific activities of the Institute of Biodiversity and Ecosystem Research-BAS. In North Macedonia, the material was collected with collaboration with the Macedonian Ecological Society (https://mes.org.mk/en/) and the Biology Students' Research Society during their field studies with the respective permissions provided. In Montenegro, Serbia, and Albania we did not need a permit for collecting for scientific purposes because it was outside protected areas, and animals were not protected.

## Geometric morphometrics

In total, 196 specimens belonging to 16 taxa of the *Poecilimon ornatus* group were used for geometric morphometric analyses. Four morphostructures (male pronotum, male cercus, ovipositor, and male tegmen) were photographed using a stereomicroscope (Leica M165C) equipped with a digital camera (Leica DMC5400) under strictly maintained magnification and resolution and saved in jpg format. TPS files for each structure were created from the photographs with the software tpsUtil v.1.26 following *Rohlf (2004)*. To explore the patterns of morphological variation, 8 landmarks (including 1 semilandmark) of male pronotum, 13 (7 semilandmarks) of male cercus, 13 (1 semilandmark) of male tegmen, and 9 (2 semilandmarks) of ovipositor (Fig. 2) were plotted manually in tpsDIG2 v.2.17 (*Rohlf, 2015*). The list of landmarks and semilandmarks used in this study is included in Table 2. After plotting the landmarks, the intersections marked in the TPS files were aligned using a Procrustes superimposition. Partial warp scores were studied using Canonical variate
**Table 1** The number of specimens used for the geometric morphometric analysis.

| Species | Male cercus | Male tegmen | Ovipositor | Male pronotum |
|---|---|---|---|---|
| *Poecilimon affinis affinis*[*] (Frivaldszky, 1868) | 29 | 26 | 11 | 23 |
| *Poecilimon affinis komareki* [*] Cejchan, 1957 | 6 | 3 | 3 | 3 |
| *Poecilimon affinis dinaricus* [*] Ingrisch & Pavićević, 2010 | 1 | 1 | 1 | 1 |
| *Poecilimon affinis serbicus* [*] Karaman, 1974 | 14 | 14 | 5 | 9 |
| *Poecilimon affinis hajlensis* [*] Karaman, 1974 | 4 | 6 | 2 | 5 |
| *Poecilimon affinis poecilus* [*] Ramme, 1951 | 15 | 12 | 5 | 4 |
| *Poecilimon rumijae*[*] Karaman, 1972 | 12 | 12 | 2 | 11 |
| *Poecilimon nonveilleri* [*] Ingrisch & Pavicevic, 2010 | 10 | 10 | 1 | 6 |
| *Poecilimon pseudornatus* [*] Ingrisch & Pavicevic, 2010 | 24 | 26 | 10 | 21 |
| *Poecilimon hoelzeli* Harz, 1966 | 6 | 6 | 3 | 6 |
| *Poecilimon jablanicensis* Chobanov & Heller, 2010 | 3 | 3 | 1 | 3 |
| *Poecilimon nobilis* Brunner von Wattenwyl, 1878 | 3 | 3 | 2 | 2 |
| *Poecilimon obesus* Brunner von Wattenwyl, 1878 | 12 | 8 | 3 | 11 |
| *Poecilimon gracilis* (Fieber, 1853) | – | – | 1 | 1 |
| *Poecilimon artedentatus* (Heller, 1984) | – | – | 2 | – |

**Notes.**
  [*]*Poecilimon affinis* complex.

analysis (CVA) for each structure in MorphoJ v.1.06d (*Klingenberg, 2011*). The first two Canonical Variables (CVs) with the greatest power to distinguish the groups were plotted in the same software. The Mahalanobis distance was measured and statistically tested using 10,000 permutation repeats.

## Stridulatory measurements

The length of the stridulatory file was measured and the number of stridulatory teeth was counted for 154 specimens from the *P. ornatus* group (9 specimens of *P. affinis ssp.*, 24 - *P. affinis affinis*, 1– *P. affinis dinaricus*, 7– *P. affinis hajlensis*, 5 – *P. affinis komareki*, 12 - *P. affinis serbicus*, 8 – *P. hoelzeli*, 3 – *P. jablanicensis*, 15 – *P. nobilis*, 10 – *P. nonveilleri*, 12 –*P. obesus*, 10 – *P. ornatus*, 29 – *P. pseudornatus*, 8 – *P. soulion*). Measurements were taken under stereomicroscope with the aid of an ocular micrometer. For measurement of the stridulatory file length, we used the distance from the first proximal (basal) to the last distal (apical)
**Table 2  List of the landmarks and semilandmarks of the pronotum, male cercus, tegmen, and ovipositor used in the geometric morphometric analysis.**

| The landmark number | Pronotum | Male cercus | Tegmen | Ovipositor |
|---|---|---|---|---|
| 1 | upper frontal part | groove left at base | most distant point | highest point at the base |
| 2 | upper part of mid groove | groove right at base | upper concave point | lowest point of the base |
| 3 | upper posterior point | most distant point at apex | most distant point | begging of teeth at the upper valve |
| 4 | lateral posterior point | opposite to 3* | most distant point | tip of upper valve |
| 5 | lower frontal part | middle measured approximately between 4 and 2* | concave side point | begging of teeth at the lower valve |
| 6 | lowest middle part | opposite to 5* | most distant point | middle between 1 and 3* |
| 7 | mid point between 4 and 6* | approximetly middle between 2 and 5* | most distant point | middle between 2 and 5* |
| 8 | begging of dark band | approximetly middle between 1 and 6* | most distant point of the lateral vein | upper point of gonangulum |
| 9 | | approximetly middle between 5 and 4* | bifurcation between veins | lower point of gonangulum |
| 10 | | approximetly middle between 6 and 3* | bifurcation between veins | |
| 11 | | upper end of black spine | bifurcation between veins | |
| 12 | | lower end of black spine | bifurcation between veins | |
| 13 | | tip of cercus | mark on the stridulatory vein between the points 3 and 10* | |

**Notes.**
*semilandmarks.

tooth. The tegmen was placed upside down so that the stridulatory file could be viewed with its proximal and distal ends being at the same level. This way, the distance between the ends was measured along the imaginary line connecting those. The total number of stridulatory teeth and the number of teeth within 2 mm at the middle of the stridulatory file were counted. Measurement data were analyzed using Principal Component Analysis (PCA) in Past 4.03 (https://www.nhm.uio.no/english/research/infrastructure/past/).

## Phylogenetic analyses

A fragment of the cytochrome c oxidase subunit I (COI) of mitochondrial DNA (mtDNA) was used to determine the phylogenetic relationship between the taxa. We aimed to construct a phylogenetic tree focusing on the species of the *P. affinis* complex. A total of 71 sequences of 14 *Poecilimon* taxa were obtained from GenBank (https://www.ncbi.nlm.nih.gov/genbank/). The DNA sequences were aligned using CodonCode Aligner 9.0.2 (https://www.codoncode.com/aligner) with default parameters. The maximum likelihood (ML) and Bayesian inference (BI) analyses were used to infer the phylogenetic relationships. The best-fit model of nucleotide substitution was determined with jModelTest2 (*Guindon & Gascuel, 2003*; *Darriba et al., 2013*). ML was performed in IQ-TREE (*Nguyen et al., 2015*), whereas BI in MrBayes 3.2. (*Ronquist et al., 2012*). For bootstrap analyses, 1,000 pseudoreplicates were generated. BI was carried out with 10,000,000 generations, with a sampling of trees every 100 generations. Likelihood values
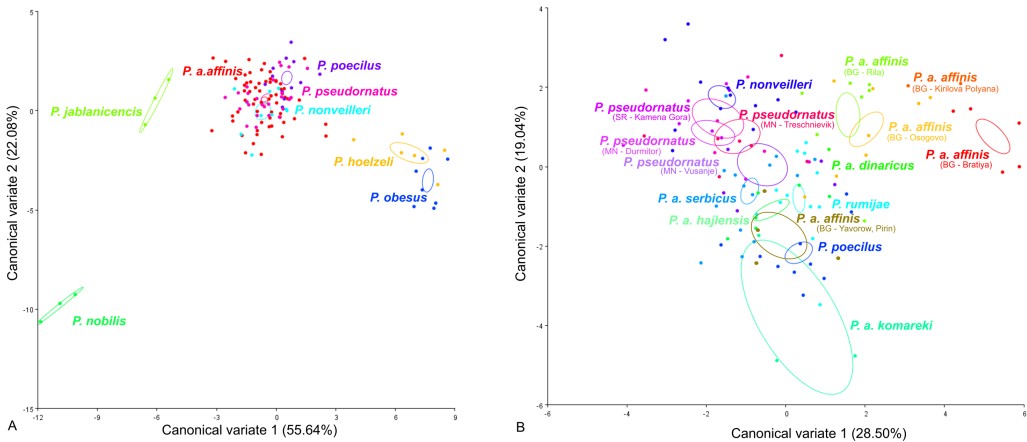

**Figure 3** **Scatter plot of the two first canonical variate axes (CV1 and CV2) analysis of centroid sizes of male tegmen:** ***P. ornatus*** **group (A) and** ***P. affinis*** **complex (B).** The different colors of the species *P. pseudornatus* and *P. a. affinis* indicate different locations from which the specimens were collected. The localities are indicated below taxa name (SR, Serbia; MN, Montenegro; BG, Bulgaria).

were observed with Tracer v.1.7 (*Rambaut et al., 2018*). The tree was visualized in FigTree 1.4.4 (*Rambaut, 2018*).

# RESULTS

## Morphology

As a result, 54 images of ovipositor, 130 of male tegmen, 142 of male pronotum, and 141 of male cercus were used in the analyses. In some specimens, tegmen and cercus were damaged and not used for this study. The landmarks were chosen based on the shape and structure of the ovipositor (seven landmarks, two semilandmarks) (Fig. 2A), male cercus (six landmarks, seven semilandmarks) (Fig. 2B), male pronotum (seven landmarks, one semilandmark) (Fig. 2C), and male tegmen (12 landmarks, one semilandmark) (Fig. 2D).

CV analysis of the male tegmen (Fig. 3) revealed significant variation within the *P. ornatus* group and *P. affinis* complex. At the species group level, the first two CV analyses together accounted for 77.72% of the total variation (CV1 = 55.64%, CV2 = 22.08%). A combination of the results of the CV1 and CV2 analyses of the male tegmen separated the species *P. hoelzeli*, *P. obesus*, *P. jablanicensis* and *P. nobilis* from the other species of the *Poecilimon ornatus* group and revealed an overlap between *P. pseudornatus*, *P. poecilus*, *P. nonveilleri*, and *P. affinis* (Fig. 3A). The Mahalanobis distance obtained through pairwise comparisons among the group revealed highly significant differences (10,000 permutation rounds; *P* < 0.0001), ranging from 2.50 (*P. affinis* and *P. pseudornatus*) to 19.66 (*P. poecilus* and *P. obesus*). The Procrustes distances also showed significant differences between groups (10,000 permutation rounds; *P* < 0.0001) ranging from 0.03 (*P. poecilus* and *P. pseudornatus*) to 0.28 (*P. nobilis* and *P. obesus*) (Table S1).

At the species complex level, the first two CVs together accounted for 47.9% of the total variation of the male tegmen (CV1 = 28.5% and CV2 = 19.4%). CV1 and CV2 analyses

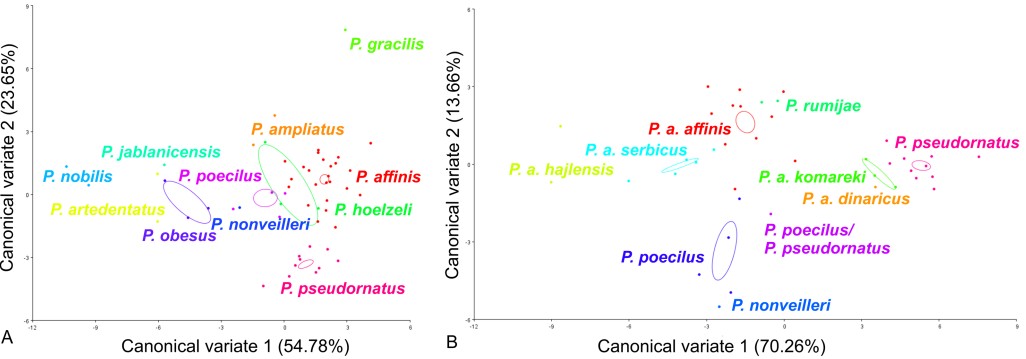

**Figure 4** **Scatter plot of the two first canonical variate axes (CV1 and CV2) analysis of centroid sizes of ovipositor: *P. ornatus* group (A) and *P. affinis* complex (B).** The different colors indicate different species/subspecies of studied bush-crickets.

of the *Poecilimon affinis* complex did not indicate clear clusters representing each of the existing species/subspecies. However, the specimens of *P. a. affinis* show differentiation in terms of their occurrence (Bratiya, Kirilova Polyana, Yavorow-Pirin, Osogovo, Rila) in contrast to *P. pseudornatus*, where specimens from different localities (Kamena Gora, Durmitor, Treschnievik, Vusanje) are grouped together (Fig. 3B). The Mahalanobis distances between taxa for male tegmen are 2.77 for *P. poecilus* and *P. pseudornatus*, and 8.13 for *P. a. komareki* and *P. a. dinaricus* (10,000 permutation rounds; $P < 0.0001$). The Procrustes distances also showed significant differences (10,000 permutation rounds; $P < 0.001$), ranging from 0.03 (*P. a. serbicus* and *P. pseudornatus*) to 0.12 (*P. rumijae* and *P. a. dinaricus*) (Table S2).

For the ovipositor, at the species group level, the first two CVs together accounted for 78.43% of the total variation (CV1 = 54.78%, CV2 = 23.65%) (Fig. 4A). The scatter plot from CV1 and CV2 shows that species from the *Poecilimon affinis* complex cannot be clearly separated from other species of the *Poecilimon ornatus* group (Fig. 4A). The Mahalanobis distances obtained by pairwise comparisons among group revealed highly significant differences (10,000 permutation rounds, $P < 0.0001$), ranging from 2.78 (*P. poecilus* and *P. hoelzeli*) to 15.72 (*P. gracilis* and *P. nobilis*). The Procrustes distances also showed significant differences between groups (10,000 permutation rounds, $P < 0.0001$) ranging from 0.04 (*P. affinis* and *P. hoelzeli*) to 0.19 (*P. pseudornatus* and *P. gracilis*) (Table S3).

At the species complex level, the first two CVs together accounted for 83.92% of the total variation of the ovipositor (CV1 = 70.26% and CV2 = 13.66%) (Fig. 4B). The centroid size (the square root of the sum of the squared distances of all landmarks from their centroid) of CV1 and CV2 shows that species from the *Poecilimon affinis* complex can be clearly separated from each other (Fig. 4B). The Mahalanobis distances obtained through pairwise comparisons of the complex revealed highly significant differences (10,000 permutation rounds; $P < 0.0001$), ranging from 2.69 (*P. rumijae* and *P. a. affinis*) to 14.50 (*P. pseudornatus* and *P. a. hajlensis*). The Procrustes distances also showed highly
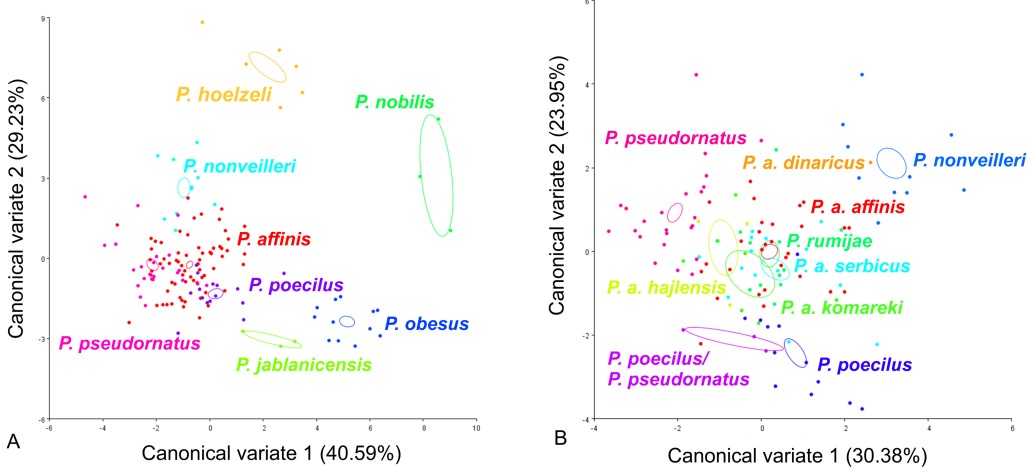

**Figure 5  Scatter plot of the two first canonical variate axes (CV1 and CV2) analysis of centroid sizes of male cercus:** ***P. ornatus*** **group (A) and** ***P. affinis*** **complex (B).** The different colors indicate different species/subspecies of the studied bush-crickets.

significant differences (10,000 permutation rounds; $P < 0,005$), ranging from 0.03 (*P. a. serbicus* and *P. a. affinis*) to 0.15 (*P. a. komareki* and *P. a. dinaricus*) (Table S4).

CV analysis of the male cercus (Fig. 5) also revealed significant variation within the *P. ornatus* group and the *P. affinis* complex. At the group level, the first two CVs together accounted for 69.82% of the total variation (CV1 = 40.59%, CV2 = 29.23%). The scatter plot from CV1 and CV2 shows that species from the *Poecilimon affinis* complex can be clearly separated from other species of the *Poecilimon ornatus* group (Fig. 5A). The Mahalanobis distances obtained through pairwise comparisons among group revealed highly significant differences (10,000 permutation rounds; $P < 0.0001$), ranging from 2.71 (*P. pseudornatus* and *P. affinis*) to 12.25 (*P. hoelzeli* and *P. jablanicensis*). The Procrustes distances also showed significant differences between groups (10,000 permutation rounds; $P < 0.0001$), ranging from 0.03 (*P. affinis* and *P. pseudornatus*) to 0.17 (*P. pseudornatus* and *P. nobilis*) (Table S5).

For the male cercus, at the complex level, the first two CVs together accounted for 54.33% of the total variation (CV1 = 30.38% and CV2 = 23.95%). The centroid size of CV1 and CV2 shows that only *P. a. affinis*, *P. rumijae*, *P. a. komareki*, and *P. nonveilleri* can be clearly separated from other members of the *P. affinis* complex (Fig. 5B). The Mahalanobis distances obtained through pairwise comparisons of the complex revealed significant differences (10,000 permutation rounds; $P < 0.0001$), ranging from 2.87 (*P. pseudornatus* and *P. a. hajlensis*) to 8.65 (*P. a. dinaricus* and *P. a. komareki*). The Procrustes distances also showed significant differences (10,000 permutation rounds; $P < 0.0001$), ranging from 0.03 (*P. a. affinis* and *P. poecilus*) to 0.10 (*P. a. komareki* and *P. nonveilleri*) (Table S6).

For the male pronotum, at the group level, the first two CVs together accounted for 75.84% of the total variation (CV1 = 57.24%, CV2 = 18,60%) (Fig. 6). The scatter plot from CV1 and CV2 shows that species from the *Poecilimon affinis* complex cannot

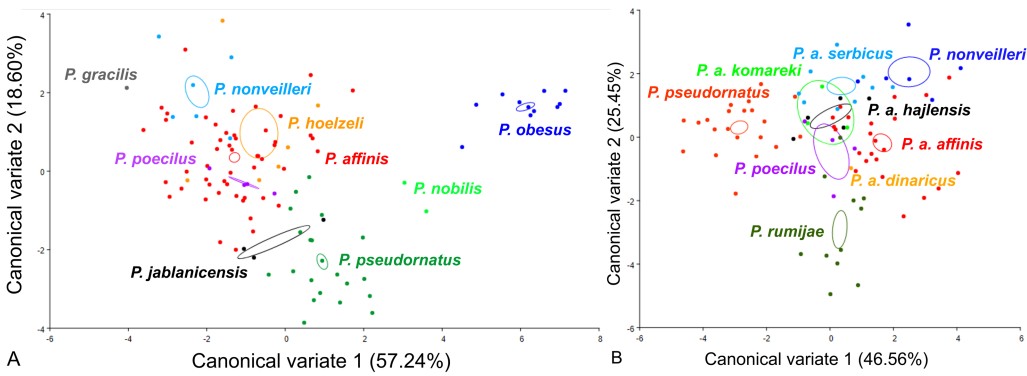

**Figure 6** Scatter plot of the two first canonical variate axes (CV1 and CV2) analysis of centroid sizes of male pronotum: *P. ornatus* group (A) and *P. affinis* complex (B). The different colors indicate different species/subspecies of the studied bush-crickets.

be clearly separated from other species of the *Poecilimon ornatus* group (Fig. 6A). The Mahalanobis distances obtained through pairwise comparisons among group revealed significant differences (10,000 permutation rounds; $P < 0.0001$), ranging from 2.20 (*P. poecilus* and *P. affinis*) to 12.81 (*P. gracilis* and *P. obesus*). The Procrustes distances also showed significant differences between groups (10,000 permutation rounds; $P < 0.0001$), ranging from 0.03 (*P. poecilus* and *P. affinis*) to 0.16 (*P. gracilis* and *P. jablanicensis*) (Table S7).

At the complex level, the first two CVs together accounted for 72.01% of the total variation of the male pronotum (CV1 = 46.56% and CV2 = 25.45%). The centroid size of CV1 and CV2 shows that only *P. rumijae* can be clearly separated from other species from the *P. affinis* complex (Fig. 6B). The Mahalanobis distances obtained through pairwise comparisons of the complex revealed significant differences (10,000 permutation rounds; $P < 0.0001$), ranging from 2.73 (*P. a. hajlensis* and *P. a. affinis*) to 5.68 (*P. rumijae* and *P. nonveilleri*). The Procrustes distances also showed highly significant differences (10,000 permutation rounds; $P < 0.0001$), ranging from 0.04 (*P. poecilus* and *P. a. affinis*) to 0.14 (*P. rumijae* and *P. nonveilleri*) (Table S8).

## Stridulatory measurements

*Poecilimon soulion* and *P. jablanicensis* have the shortest stridulatory file of all studied species (2.74–3.17 and 2.96–3.04, respectively). In contrast, *P. affinis komareki* has the longest stridulatory file (5.34–5.88) and the greatest number of teeth on its structure (158–195). *Poecilimon obesus* has the lowest number of teeth, which proves that the length of the stridulatory file does not correlate with the number of teeth (Table 3). Principal Component Analysis of the stridulatory file and the number of teeth shows that *P. nonveilleri*, *P. ornatus*, *P. hoelzeli*, *P. pseudornatus*, *P. a. serbicus*, *P. a. hajlensis*, and *P. a. affinis* overlap. Moreover, we can conclude that *P. a. affinis* is the most diverse taxon within the *P. ornatus* group, while *P. a. komareki* is the most distinct taxon of the studied group (Fig. 7).

**Table 3** Measurements for stridulatory files of the *P. ornatus* group. Measurements are given in mm: first row – min-max values; in brackets – avarage ± Standard deviation.

| Species | Number of specimens | Stridulatory length | Number of stridulatory teeth |
|---|---|---|---|
| *P. affinis* | 9 | 3.68–4.46 (4.08) | 122–169 (146) |
| *P. affinis affinis* | 24 | 3.84–4.46 (4.17 ± 0.19) | 119–151 (138 ± 12) |
| *P. affinis hajlensis* | 7 | 4.08–4.46 (4.38 ± 0.14) | 133–153 (149 ± 7) |
| *P. affinis komareki* | 5 | 5.34–5.88 (5.64 ± 0.25) | 158–195 (181 ± 15) |
| *P. affinis serbicus* | 12 | 3.84–4.37 (4.14 ± 0.21) | 136–156 (144 ± 6) |
| *P. hoelzeli* | 8 | 4.14–5.34 (4.85 ± 0.42) | 125–150 (141 ± 8) |
| *P. jablanicensis* | 3 | 2.96–3.04 (3.01 ± 0.05) | 121–135 (128 ± 7) |
| *P. nobilis* | 15 | 2.78–3.98 (3.28 ± 0.33) | 81–111 (97 ± 9) |
| *P. nonveilleri* | 10 | 3.74–4.32 (3.97 ± 0.18) | 104–119 (111 ± 5) |
| *P. obesus* | 12 | 3.37–4.6 (4.28 ± 0.31) | 80–110 (92 ± 8) |
| *P. ornatus* | 10 | 3.74–4.6 (4.08 ± 0.31) | 105–128 (117 ± 7) |
| *P. pseudorantus* | 29 | 4.22–4.9 (4.66 ± 0.16) | 125–147 (139 ± 5) |
| *P. soulion* | 8 | 2.74–3.17 (2.99 ± 0.13) | 97–103 (99 ± 2) |
| *P. affinis dinaricus* | 1 | 5.38 | 149 |
| *P. artedentatus* | 1 | 4.8 | 168 |

## Phylogenetic analyses

The final alignment consists of 607 bp, of which 450 were conservative, 157 variable and 83 parsimony-informative sites. HKY+G was selected as the best-fit evolution model for site substitution. The topologies obtained from BI and ML analyses were similar. Bootstrap values (ML) (>50%) and BI posterior probabilities (>0.5) are shown on the nodes of the tree presented on Fig. 8. To root the tree, *Poecilimon cervus* Karabag, 1950, belonging to the *Poecilimon bosphoricus* Brunner von Wattenwyl, 1878 species group, was chosen. The BI and ML trees based on the COI data show that the *P. affinis* complex forms a paraphyletic group. The most diverse taxon in the complex is *P. a. affinis*, occupying different nodes on the phylogenetic tree grouping by geographic locality. *Poecilimon a. affinis* from Kirilova Polyana (Bulgaria, Rila Mtns) occupies a basal position in the tree and seems to be a sister taxon to the remaining taxa of the complex. Two species of the *P. ornatus* group, preliminary left outside the *P. affinis* complex, *P. ornatus* and *P. hoelzeli,* were placed within the same clade (Fig. 8).

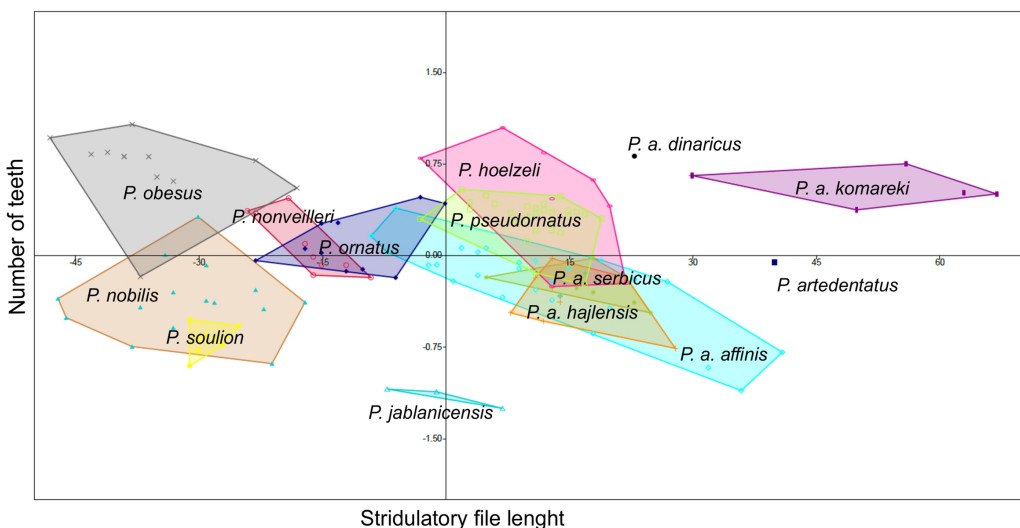

**Figure 7** **Principal Component Analysis (PCA) of stridulatory measurements and number of teeth:** ***P. ornatus* group.** The different colors indicate different species/subspecies of studied bush-crickets.

## DISCUSSION

### Morphology

This work aimed to determine the morphological characteristics that separate bush-crickets belonging to the *P. affinis* complex from other species of the *P. ornatus* group through the geometric morphometrics approach. The morphology of the male tegmen, ovipositor, cercus and male pronotum were used successfully in morphological studies of *Poecilimon* (*Heller, 2004*; *Chobanov & Heller, 2010*; *Kaya et al., 2012*; *Kaya, Boztepe & Çiplak, 2015*; *Kaya et al., 2018*). The present work showed that the studied morphostructures can partly be used to separate taxa of the species rank in the *Poecilimon ornatus* group. *Chobanov & Heller (2010)* noticed that the pronotal shape and the size of the area of the male tegmen covered by the male pronotum vary between specimens from the same locality. Our results support the poor taxonomic utility of the shape of male pronotum in this group for distinguishing the species belonging to the *P. affinis* complex from other species in the group (Fig. 6A). However, based on the shape of the male tegmen, *P. affinis* and its subspecies group with *P. nonveilleri*, *P. pseudornatus* in the same place, which clearly separates them from other species (Fig. 3A). This may confirm our assumption for the designation of the *P. affinis* complex including other species from the *Poecilimon ornatus* group. CV analysis of centroid sizes of the male pronotum (Fig. 6B) shows that *P. rumijae* is the most distinct taxon among the *P. affinis* complex, and does not overlap with *P. a. komareki*. *Poecilimon rumijae* may likely be treated as a separate species of the *P. ornatus* group, differing distinctly from subspecies of *P. affinis* (*Ingrisch & Pavicevic, 2010*), but further studies are required to confirm its taxonomic position. This assumption is also confirmed by the analysis of the ovipositor, where *P. a. komareki* is more similar to *P. a. dinaricus* and *P. pseudornatus*, whereas *P. rumijae* is more similar to *P. a. affinis* (Fig. 4B).

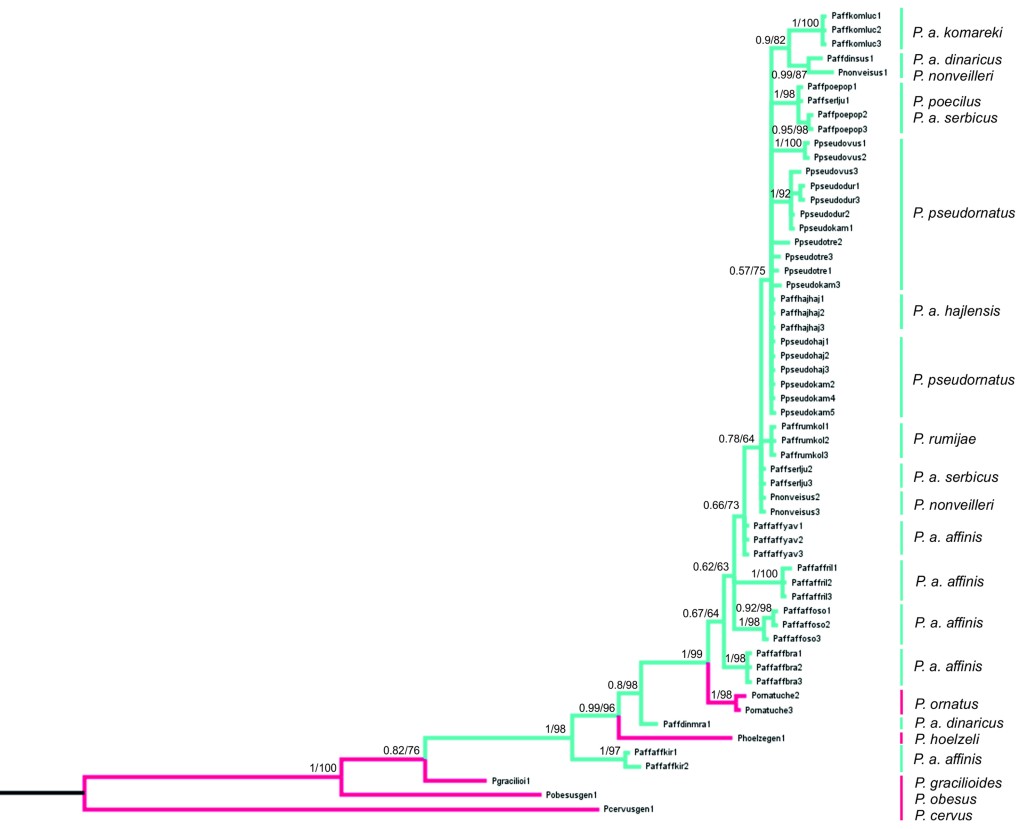

**Figure 8** **Phylogenetic tree based on Bayesian inference method including COI sequences of the *P. affinis* complex.** Bayesian Inference (BI) and Maximum Likelihood (ML) topologies were similar, so only one tree is shown. BI posterior probability (<0.5) and ML bootstrap values (<50) are shown on the nodes. Pink indicates species outside of the *P. affinis* complex; green indicates taxa from the designated *P. affinis* complex.

On the other hand, the results based on the male cercus (Fig. 5B) show that *P. a. komareki* and *P. rumijae* overlap, which proves high similarities within this morphostructure and may confirm the accuracy of lowering *P. rumijae* to the rank synonymous with *P. a. komareki* (*Chobanov & Heller, 2010*; *Cigliano et al., 2021*). *Ingrisch & Pavicevic (2010)* considered *P. rumijae* to be similar to *P. nonveilleri* and *P. affinis*. Our results confirm a close relationship between *P. rumijae* and *P. affinis*, but not between *P. rumijae* and *P. nonveilleri*, which, according to all morphostructures, are the most distant from each other (Figs. 3A, 4A, 5A, 6A).

The most distinct species in our sample is *P. nobilis* based on the analysis of the male tegmen (Fig. 3A) and male cercus (Fig. 5A), *P. gracilis* based on the ovipositor (Fig. 4A), and *P. obesus* based on the male pronotum (Fig. 6A), which suggest not to include these species in the *P. affinis* complex. On the other hand, *P. affinis* is the most diffuse taxon in the group (Figs. 3A, 4A, 5A, 6A). The results suggest that the difference between specimens of *P. a. affinis* is related to the locality in which they occur (Fig. 3B), and is generally connected with altitude (*Chobanov & Heller, 2010*). Specimens of *P. a. affinis* from Pirin are distant

from individuals from Bratiya, Kirilowa Polyana, Osogovo, Rila and are more closely related to *P. poecilus*, *P. a. hajlensis* and *P. a. komareki* (Fig. 3B). On the other hand, the position of the centroid size of *P. pseudornatus* from different localities (Durmitor, Kamena Gora, Treshnievik, Vusanje) overlaps, which proves a lower morphological variability in terms of location than in the case of *P. a. affinis* (Fig. 3B). At the group level, based on the male cercus (Fig. 5A), species from the *P. affinis* complex (*P. affinis* with its subspecies, *P. nonveilleri* and *P. pseudornatus*) overlap. Thus, this is the second morphostructure to confirm the existence of this complex. Additionally, *Chobanov & Heller (2010)* suggested that the male cercus may be a better feature for separating species in this group. The results of the CV analysis of centroid size of the ovipositor (Fig. 4A) show the similarity between *P. affinis*, *P. hoelzeli*, *P. pseudornatus*, *P. poecilus*, and *P. nonveilleri*, which may indicate the extension of the *P. affinis* complex with *P. hoelzeli* (Fig. 4A). *Poecilimon poecilus*, which we suggested to treat separately in this work, seems to fell within the variation of *P. a. affinis*. It is confirmed by all the morphostructures studied, where *P. poecilus* overlaps with other subspecies: *P. a. affinis, P. a. hajlensis, P. a. komareki* (Figs. 3A, 4A, 5A, 6A). However, to establish the taxonomic status of *P. poecilus*, additional research is needed.

## Stridulatory structures measurements

The stridulatory file and the number of teeth can be a good morphological feature for distinguishing taxa in the *P. ornatus* group (*Heller, 1984*; *Willemse, 1985*; *Heller, 1988*; *Chobanov & Heller, 2010*). *Heller (1988)* reports that *P. ornatus* has fewer teeth than *P. affinis,* about 158–212, with some exceptions of large specimens having up to 220 teeth, as confirmed by our results (Table 3). The length of stridulatory file is the same in both species and averaged 4.08. Thus, this morphostructure and the number of teeth are not a good feature for distinguishing *P. affinis* from *P. ornatus*. *Heller (1984)* observed about 220–230 teeth in *P. affinis* species, while *Chobanov & Heller (2010)* observed 180–240. They suggest that the number is generally more variable in southeastern populations (SW Bulgaria). The lowest number of teeth is found in small specimens from high altitudes. Principal Component Analysis (PCA) shows a similarity between three subspecies (*P. a. affinis, P. a. serbicus* and *P. a. hajlensis*) (Fig. 7). On the other hand, *P. a. komareki* does not overlap with other subspecies, which may mean that it is the most distinct taxon from all studied taxa of the *P. ornatus* group. *Poecilimon hoelzeli* and *P. pseudornatus* have a similar number of teeth and length of the stridulatory file. *Poecilimon ornatus*, *P. nonveilleri*, *P. a. affinis*, *P. a. hajlensis*, *P. a. serbicus*, *P. pseudornatus* and *P. hoelzeli* overlap, which can suggest that *P. hoelzeli* and *P. ornatus* should be included in the designated *P. affinis* complex.

## Phylogenetic data

The first genetic studies using ribosomal internal transcribed spacers (ITS1 and 2) and the mitochondrial genes (16S rRNA, tRNA-Val, 12S rRNA) involving some of the group's species were conducted by *Ullrich et al. (2010)*. However, they did not provide conclusive information on the relationship between species in this group. *Kociński (2020)* performed a genetic analysis based on the cytochrome c oxidase I gene (COI) of the *P. ornatus* group, and confirmed the monophyly of this group. Our results, focusing on species from

the *P. affinis* complex, show that it forms a paraphyletic group (Fig. 8). Two additional species, *P. hoelzeli* and *P. ornatus*, are distributed with the other taxa of the complex, thus they probably should be included in the *P. affinis* complex determined previously. This assumption is similar to the results of the CVA of the ovipositor, where taxa from the complex overlap with *P. hoelzeli* (Fig. 4A). Moreover, based on the phylogenetic tree (Fig. 8), *P. a. affinis* is the most diverse species in the complex, occupying different nodes, which is supported by the CVA results of the male tegmen (Fig. 3B). The variability is related to the location (Bratiya, Kirilova Polyana, Rila, Yavorow) of the populations of *P. a. affinis*, and is connected with the altitude of occurrence (*Chobanov & Heller, 2010*). *Poecilimon a. komareki* and *P. rumijae* form different nodes, which may suggest treating them as separate taxa of the *P. ornatus* group. This opinion is confirmed by the CVA results of male pronotum and ovipositor (Figs. 4B, 6B). The specimens from *P. poecilus* also form different nodes compared to *P. a. affinis*, thus, it may be treated as a subspecies of *P. affinis*, which is supported by the CVA of the male tegmen, male cercus, ovipositor, and male pronotum (Figs. 3B, 4B, 5B, 6B).

## CONCLUSIONS

The geometric morphometric method has proven to be useful in studying the morphological diversity of bush-crickets. Combined with the analysis of the stridulatory file and molecular phylogeny, it provides better insight into the relationships between species from the *Poecilimon ornatus* group, and in particular, the taxa of the *Poecilimon affinis* complex. Morphological analysis of selected morphostructures and molecular data showed the paraphyly of the *P. affinis* complex unless *P. ornatus* and *P. hoelzeli* are included. Additionally, the taxonomic status of *P. rumijae* and *P. poecilus* remains unclear. Our results show some discordances with previous studies and point to the need for a most thorough interdisciplinary phenetic and genetic study in order to solve the systematics of this particular group of bush-crickets.

## ACKNOWLEDGEMENTS

We thank the Biology Students' Research Society (BSRS; Skopje, Republic of North Macedonia) and its 2017 chair, Marija Trencheva, for the accommodation and logistic support, and Slobodan Ivković for the help in the field, during our collecting trips in Macedonia.

### Funding

This work was supported by the National Science Fund (MES) of Bulgaria to Dragan Chobanov (DN11/14–18.12.2017) and a Bilateral Agreement between the Polish and Bulgarian Academies of Sciences (project: Convergent evolution of polyphyletic bush-crickets (Orthoptera: Phaneropterinae): micropterism and speciation). The funders had no role in study design, data collection and analysis, decision to publish, or preparation of the manuscript.

## Grant Disclosures

The following grant information was disclosed by the authors:

The National Science Fund (MES) of Bulgaria to Dragan Chobanov: DN11/14–18.12.2017.

A Bilateral Agreement between the Polish and Bulgarian Academies of Sciences.

## Competing Interests

The authors declare there are no competing interests.

## Author Contributions

- Maciej Kociński conceived and designed the experiments, performed the experiments, analyzed the data, prepared figures and/or tables, authored or reviewed drafts of the paper, and approved the final draft.
- Beata Grzywacz performed the experiments, authored or reviewed drafts of the paper, and approved the final draft.
- Georgi Hristov analyzed the data, authored or reviewed drafts of the paper, and approved the final draft.
- Dragan Chobanov conceived and designed the experiments, authored or reviewed drafts of the paper, and approved the final draft.

## Field Study Permissions

The following information was supplied relating to field study approvals (i.e., approving body and any reference numbers):

In Greece, field studies were approved by the Greek Ministry of the Environment, Energy, and Climate Change.

In Bulgaria, we did not need a permit for collecting for scientific purposes because it was outside protected areas, and animals were not protected. The material was collected with scientific purpose through scientific activities of the Institute of Biodiversity and Ecosystem Research-BAS. In North Macedonia, the material was collected with collaboration with the Macedonian Ecological Society (https://mes.org.mk/en/) and the Biology Students' Research Society during their field studies with the respective permissions provided. In Montenegro and Albania, we did not need a permit for collecting for scientific purposes because it was outside protected areas, and animals were not protected.

In Serbia, we also did not need a permit for collecting insects because it was outside protected areas, and animals were not protected.

## Data Availability

The 4 morphostructures (male tegmen, ovipositor, cercus, male pronotum) and the localization of each landmark of each specimen and the measurements of the stridulatory's file and the number of teeth of each specimen are available in the Supplementary Files.

## Supplemental Information

Supplemental information for this article can be found online at http://dx.doi.org/10.7717/peerj.12668#supplemental-information.

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
