# Peer review of "A taxonomic outline of the Poecilimon affinis complex (Orthoptera) using the geometric morphometric approach"

_PeerJ, doi:10.7717/peerj.12668_

## Round 0.1 · original submission · Minor Revisions

The reviewers have suggested some minor corrections. Please make them and send back the corrected version asap.

·

Basic reporting

The paper is clearly written and much original.
Only a few small suggestions are reported in the attached text.

Experimental design

Original and well defined.

Validity of the findings

Findings show that the method followed by the authors is good.

Additional comments

Further, I would suggest to the authors to make a decision about some species: are they synonyms or valid species?

Reviewer 2 ·

Basic reporting

The English language is clear and professional throughout the text, the Introduction and
background show the general context however I believe it is important to include the differences
between P. rumijae and P. affinis. The literature citation is appropriate for this manuscript and the
figures are relevant and well labeled and described. The structure is in accordance with PeerJ.

Experimental design

This manuscript A taxonomic outline of the Poecilimon affinis complex (Orthoptera) using the
geometric morphometric approach, represents a major contribution, especially by including a the
geometric morphometric analysis study to explore variation in the structure of the tegmen,
ovipositor, cercus and pronotum.

Below, I present some considerations about the manuscript.

1.- In the part of the methods on geometric morphometric analysis, I consider that they should be
clearer in the placement of landmarks so that these can be replicated.
2.- Why only use landmarks and not semilandmarks?

Validity of the findings

It appears that all the data are robust and statistically sound, but until it is clear how the
decision to place each landmarks was made, it will not be sufficiently clear.

Annotated reviews are not available for download in order to protect the identity of reviewers who chose to remain anonymous.

·

Basic reporting

no comment

Experimental design

no comment

Validity of the findings

no comment

Additional comments

I am not really familiar to the methods used in this study, but the authors seem to know well what they are doing. Below a few suggestions/comments with line numbers:

35 According to OSF (Cigliano et al.) the most species-rich phaneropterine genus is Anaulacomera from the Neotropics (154 valid species, Poecilimon has 145).

86 cercus > male cercus

183 complex species > species complex?

289/290 it occurs > they occur

343 than > compared to?

344 its subspecies – it cannot be a subspecies of a subspecies (P. a. affinis)

354 ff. The last sentence is not very clear.

Figure 1: Photo > Photos

Figure 3: The figures are very small. It is almost impossible to read the species names and the even smaller locality names below them without greatly zooming in. It would probably be better to enlarge both figures to full page width.

Figure 4: Same here, here, species names not as tyny as in Fig. 3 but still very small (and yellow is almost imposiible to read).

Figures 5,6: Same as in 4.

---

## Round 0.2 · accepted · Accept

Thanks for revising the manuscript following the reviewer's suggestions.